# Short communication: Distribution of phospholipids in parotid cancer by matrix-assisted laser desorption/ionization imaging mass spectrometry

Hirofumi Kanetake[1]*, Nahoko Kato-Kogoe[2], Tetsuya Terada[1], Yoshitaka Kurisu[3], Wataru Hamada[2], Yoichiro Nakajima[2], Yoshinobu Hirose[3], Takaaki Ueno[2], Ryo Kawata[1]

1 Department of Otorhinolaryngology-Head and Neck Surgery, Faculty of Medicine, Osaka Medical and Pharmaceutical University, Takatsuki, Japan, 2 Department of Dentistry and Oral Surgery, Faculty of Medicine, Osaka Medical and Pharmaceutical University, Takatsuki, Japan, 3 Department of Pathology, Faculty of Medicine, Osaka Medical and Pharmaceutical University, Takatsuki, Japan

* hirofumi.kanetake@ompu.ac.jp

**Data Availability Statement:** All relevant data are within the manuscript and its S1–S3 Figs files.

## Abstract

### Background

Parotid cancer is relatively rare, and malignancy varies; therefore, novel markers are needed to predict prognosis. Recent advances in matrix-assisted laser desorption/ionization imaging mass spectrometry (MALDI-IMS), useful for visualization of lipid molecules, have revealed the relationship between cancer and lipid metabolism, indicating the potential of lipids as biomarkers. However, the distribution and importance of phospholipids in parotid cancer remain unclear.

### Objective

This study aimed to use MALDI-IMS to comprehensively investigate the spatial distribution of phospholipids characteristically expressed in human parotid cancer tissues.

### Methods

Tissue samples were surgically collected from two patients with parotid cancer (acinic cell carcinoma and mucoepidermoid carcinoma). Frozen sections of the samples were assessed using MALDI-IMS in both positive and negative ion modes, with an *m/z* range of 600–1000. The mass spectra obtained in the tumor and non-tumor regions were compared and analyzed. Ion images corresponding to the peak characteristics of the tumor regions were visualized.

### Results

Several candidate phospholipids with significantly different expression levels were detected between the tumor and non-tumor regions. The number of unique lipid peaks with significantly different intensities between the tumor and non-tumor regions was 95 and 85 for

**Funding:** The funders had no role in study design, data collection and analysis, decision to publish, and preparations of the manuscript.

**Competing interests:** The authors have declared that no competing interests exist.

Cases 1 and 2, respectively, in positive ion mode, and 99 and 97 for Cases 1 and 2, respectively, in negative ion mode. Imaging differentiated the characteristics that phospholipids were heterogeneously distributed in the tumor regions.

## Conclusion

Phospholipid candidates that are characteristically expressed in human parotid cancer tissues were found, demonstrating the localization of their expression. These findings are notable for further investigation of alterations in lipid metabolism of parotid cancer and may have potential for the development of phospholipids as biomarkers.

## Introduction

Parotid cancer, the most common salivary gland cancer, has various histological types and grades of malignancy [1, 2]. Because of its clinical and histological variety, it is difficult to estimate the malignancy grade and histological type preoperatively [3]. Therefore, not only the improvement of clinical and pathological diagnosis, but also prognostic estimation methods for parotid cancer are necessary to provide appropriate treatment according to the grade of malignancy. In addition, the carcinogenic mechanism of parotid cancer remains unclear because of its rarity and variety of histological types. Although specific genes and proteins have been evaluated as practical markers for parotid cancer, lipid molecules have not been investigated to date [4–6].

Lipids play an important role in various biological functions [7]. In recent years, the role of lipid molecules in biological functions has been clarified, and the importance of the relationship between cancer and lipid metabolism has attracted much attention [8–10]. Since lipid metabolism affects cellular processes, such as cell growth and division, and is associated with carcinogenesis, it has been suggested that phospholipid-related compounds may be new biomarkers [11].

Matrix-assisted laser desorption/ionization imaging mass spectrometry (MALDI-IMS) technology has recently been advanced to investigate the distribution of phospholipid expression [12]. Compared to conventional methods that require labeling, mass spectrometry (MS) allows the analysis of a wide variety of molecules, with only minor structural differences and without the need for labeling. In addition, imaging mass spectrometry (IMS) enables visualization of the distribution of many biomolecules by overlaying microscopic images and mass spectrometric data for image analysis [13, 14]. MALDI-IMS has been used to study the localization of proteins and phospholipids in some cancer tissues [15, 16]. In preceding studies, the expression patterns of specific phospholipids have been reported to differ between cancerous and non-cancerous regions in adenocarcinomas such as prostate and breast cancers [17, 18]. Furthermore, it has been suggested that these phospholipids may be potential biomarkers [15, 19, 20]. Therefore, it is possible that the expression of phospholipids is also a characteristic of the difference between cancerous and non-cancerous regions in parotid cancer, but there have been no reports on phospholipids in parotid cancer.

In this study, a MALDI-IMS-based lipidomic strategy was employed to profile the differentially expressed candidate phospholipids between parotid cancer tissues and the corresponding adjacent non-cancerous salivary gland tissues as a first step toward finding the importance of phospholipids in parotid cancer.

## Materials and methods

### Ethics statement

The present study was conducted in accordance with the Declaration of Helsinki and its latest amendments, and was approved by the Ethics Committee of Osaka Medical College (approval no. 2020–211). Written informed consent was obtained from all participants.

### Sample collection

Tissue samples were collected from surgically removed tissues of two Japanese patients who underwent surgery with a diagnosis of parotid cancer at Osaka Medical College Hospital, Takatsuki City, Japan, in 2020. The patients were not previously diagnosed and had not received any prior treatment. They were both male, 32 and 65 years of age. The histologic type of cancer tissue was acinic cell carcinoma and mucoepidermoid carcinoma, and both were low- to intermediate-grade malignancies. Tissue samples were cut into tissue blocks immediately after surgical removal (S1 Fig), frozen in isopentane cooled to -80˚C with dry ice, and stored in a -80˚C freezer until analysis.

### Tissue section preparation

The tissue blocks were sliced to a thickness of 10 μm at -20˚C using a cryostat (CM1950; Leica, Wetzler, Germany). Serial tissue sections were mounted on an indium tin oxide-coated glass slide (Bruker Daltonics, Bremen, Germany) and a Matsunami adhesive silane-coated glass slide (Matsunami, Osaka, Japan) for IMS analysis and hematoxylin and eosin (HE) staining, respectively. Before analysis by IMS, the pathologist confirmed that the tumor and non-tumor regions of parotid cancer were present in a single section using HE-stained images of serial tissue sections. The non-tumor areas confirmed to be normal parotid tissue were included in the analysis. Regions with insufficient pathological findings were excluded from the analysis.

For IMS, each tissue section was coated with 9-aminoacridine (Merck, Darmstadt, Germany), which served as the matrix. Each slide was coated with a 9-aminoacridine matrix layer obtained by sublimation at 220˚C, and the thickness was set to 1 μm using IMLayer (Shimadzu Corporation, Kyoto, Japan).

### IMS analysis

The tissue sections were analyzed using an imaging mass microscope (iMScope TRIO, Shimadzu Corporation, Kyoto, Japan) equipped with a 355-nm Nd: YAG laser, which has a mass resolution of 10,000 and a mass accuracy of less than 20 ppm. Mass spectrometry data were acquired in positive and negative ion modes in the 600.0–1000.0 $m/z$ range using an external calibration method. The interval between each data point was 50 μm. The mass spectrometry parameters were manually optimized to obtain the highest sensitivity.

### IMS data analysis

Imaging MS Solution version 1.12.26 (Shimadzu Corporation, Kyoto, Japan) was used to normalize the mass data to the total ion current and eliminate variations in ionization efficiency. A total of 100 spectra with strong average intensities were extracted, and hierarchical cluster analysis was performed using the Euclidean distance analysis method for inter-individual distances and Ward's method for cluster distances. Of the cluster images obtained by hierarchical cluster analysis, mass spectrometry images of the masses specifically detected in the tissue were created. The phospholipids analyzed were phosphatidylcholine (PC), phosphatidylethanolamine (PE), phosphatidylinositol (PI), phosphatidylserine (PS), phosphatidylglycerol (PG),

and phosphatidic acid (PA), along with sphingomyelin (SM). Candidate compounds with masses (*m/z*) consistent with these theoretical masses (assumed to be [M+H]$^+$ or [M-H]$^-$) were estimated from the Human Metabolome Database (https://hmdb.ca/spectra/ms/search).

Based on the results of the HE staining of serial sections, regions of interest (ROIs) were defined for parotid cancer areas and non-tumor areas (S1C and S1D Fig). The signal intensities of each defined ROI were statistically compared using the Welch's t-test. A p-value of < 0.01 was set to be statistically significant.

## Results

### Histopathological findings

The pathological diagnosis of Case 1 was acinic cell carcinoma, and that of Case 2 was mucoepidermoid carcinoma. Both were cases of locally advanced intermediate-grade cancer. The tissue removed during surgery in Case 1 was approximately 4 cm in diameter, and the cut surface of the tumor was yellowish-brown and cystic (S1A Fig). HE staining in Case 1 revealed serous glandular cell-like tumor cells with hematoxylin-stained basophilic granules in the cytoplasm, which proliferated in a solid pattern (S1C Fig). In Case 2, the tumor was approximately 4 cm in diameter, the cut surface was bifid, and numerous cystic spaces filled with mucus were observed (S1B Fig). The enlarged image revealed presence of mucous cells, epidermoid cells, and intermediate cells (S1D Fig). The ROI areas of the tissue corresponding to the tumor and non-tumor areas were determined by an experienced pathologist.

### Differences in mass spectra of tumor and non-tumor region

Two parotid cancer tissue specimens from the two cases were analyzed by IMS. Those contained tumor and non-tumor regions in the same section. The average mass spectra obtained from the tumor and non-tumor ROIs determined by HE staining in S1 Fig are shown in S2 and S3 Figs. The number of peaks with significantly different intensities (Welch's t-test, p < 0.01) between the two ROIs were 95 and 99 in positive and negative ion modes for Case 1, respectively, and 85 and 97 in positive and negative ion modes for Case 2, respectively.

Among these, altered *m/z* values for which the ratio of the average spectral intensities between ROIs was more than double, and the median value of spectral intensities is more than 200 in the positive ion mode, identified by IMS in positive ion mode and negative ion mode were shown in Tables 1 and 2, respectively. The numbers of *m/z* common in Case 1 and 2 in the positive ion mode were three (*m/z* 705.58, 706.53, and 725.56) with increased expression in tumors and seven (*m/z* 786.61, 787.62, 788.61, 952.66, 953.66, 954.68, and 980.70) with decreased expression in tumors (Table 1); whereas, in the negative ion mode, there were five (*m/z* 616.45, 630.46, 642.46, 659.16, and 661.46) with increased expression in tumors and four (*m/z* 770.51, 786.50, 861.52, and 862.52) with decreased expression in tumors (Table 2).

### Visualization by IMS of phospholipid candidates in human parotid cancer tissues

Among the images of each *m/z* obtained by IMS, Figs 1A and 2A show representative images of peaks (*m/z*) with significantly different signal intensities in the tumor and non-tumor areas, which are indicated by red and blue circles on the tumor/non-tumor dot graph in S2 and S3 Figs. Each *m/z* clearly distinguished the difference between tumor and non-tumor areas. Its expression was observed to be heterogeneously distributed within the tumor and non-tumor areas, respectively. The signal intensities of the tumor area (T) and non-tumor area (NT) at each *m/z* were significantly different (Welch's t-test, p < 0.01) (Figs 1B and 2B).

**Table 1. Altered peaks identified by MALDI-IMS and corresponding candidate phospholipids in positive ion mode.**

**A.** Up-regulated in Tumor

| Case 1 | | | | | Case 2 | | | | |
|---|---|---|---|---|---|---|---|---|---|
| m/z | ratio[a] | Candidate phospholipids | | | m/z | ratio[a] | Candidate phospholipids | | |
| 706.53 | 4.3 | PC(30:0) | PE(33:0) | | 666.49 | 3.2 | - | | |
| 725.56 | 3.4 | PA(38:4) | | | 697.48 | 3.2 | PA(34:1) | PA(36:4) | PS(28:0) |
| 726.56 | 3.2 | PE(36:3)* | | | | | PE(33:4) | PG(30:2) | |
| 741.53 | 2.9 | - | | | 667.50 | 3.0 | PG(28:0) | | |
| 703.59 | 2.6 | SM(34:1) | | | 727.58 | 2.7 | - | | |
| 704.59 | 2.5 | PC(31:0)* | PE(34:1)* | PE(34:0)* | 706.56 | 2.4 | PC(30:0) | PE(33:0) | |
| 720.58 | 2.4 | PC(31:0) | PE(34:0) | | 698.48 | 2.3 | PE(33:4) | PE(34:3)* | |
| 705.58 | 2.4 | SM(34:0) | PA(36:0) | | 694.47 | 2.2 | PS(29:0) | | |
| 734.57 | 2.2 | PC(32:0) | PE(35:0) | | 705.60 | 2.2 | SM(34:0) | | |
| 735.57 | 2.1 | - | | | 688.41 | 2.1 | - | | |
| | | | | | 725.56 | 2.0 | PA(38:4) | | |

**B.** Down-regulated in Tumor

| Case 1 | | | | | Case 2 | | | | |
|---|---|---|---|---|---|---|---|---|---|
| m/z | ratio[a] | Candidate phospholipids | | | m/z | ratio[a] | Candidate phospholipids | | |
| 860.53 | 0.2 | PS(42:8) | | | 796.54 | 0.2 | PS(37:5) | PE(40:4) | PC(37:4) |
| 980.70 | 0.3 | - | | | 759.58 | 0.3 | PA(40:1) | | |
| 786.61 | 0.4 | PC(36:2) | PE(40:1)* | | 953.66 | 0.3 | - | | |
| 953.66 | 0.4 | - | | | 952.66 | 0.3 | - | | |
| 787.62 | 0.4 | PA(42:1) | SM(38:1) | | 798.56 | 0.3 | PS(37:4) | PE(40:3) | |
| 952.66 | 0.4 | - | | | 785.60 | 0.4 | PA(42:2) | | |
| 955.68 | 0.4 | - | | | 786.61 | 0.4 | PC(36:2) | PE(40:1)* | |
| 954.68 | 0.4 | - | | | 784.59 | 0.4 | PC(36:3) | PE(40:2)* | |
| 788.61 | 0.5 | PE(39:1) | PC(36:1) | PE(40:0)* | 787.62 | 0.4 | PA(42:1) | | |
| 675.47 | 0.5 | PA(34:1) | | | 758.59 | 0.4 | PE(37:2) | PC(34:2) | PE(38:1)* |
| | | | | | 780.55 | 0.4 | PC(36:5) | PE(40:4)* | |
| | | | | | 804.55 | 0.4 | PC(38:7) | PS(36:1) | |
| | | | | | 781.56 | 0.4 | PA(42:4) | | |
| | | | | | 806.57 | 0.4 | PC(38:6) | PS(36:0) | |
| | | | | | 807.57 | 0.4 | PA(44:5) | PG(38:0) | |
| | | | | | 980.70 | 0.4 | - | | |
| | | | | | 731.61 | 0.4 | SM(36:1) | | |
| | | | | | 783.58 | 0.5 | PA(42:3) | | |
| | | | | | 788.61 | 0.5 | PE(39:1) | PC(36:1) | PE(40:0)* |
| | | | | | 782.57 | 0.5 | PC(36:4) | PE(40:3)* | |
| | | | | | 760.60 | 0.5 | PC(34:1) | PE(37:1) | PE(38:0)* |
| | | | | | 761.60 | 0.5 | PA(40:0) | | |
| | | | | | 808.59 | 0.5 | PC(38:5) | | |
| | | | | | 809.59 | 0.5 | PA(44:4) | | |
| | | | | | 789.62 | 0.5 | PA(42:0) | | |
| | | | | | 954.68 | 0.5 | - | | |

The list of *m/z*, where the mean spectrum is significantly different between tumor and non-tumor regions (Welch's t-test, p < 0.01) and the ratio of mean spectral intensities is more than double, and the median value of spectral intensities is more than 200. The candidate phospholipids estimated from the Human Metabolome Database (https://hmdb.ca/spectra/ms/search)corresponding to these *m/z* values are presented. The *m/z* common to Case 1 and Case 2 is shown by shading.

[a] The ratio of relative signal intensity of tumor region to non-tumor region.

*deoxidation products. PC: phosphatidylcholine, PE: phosphatidylethanolamine, PI: phosphatidylinositol, PS: phosphatidylserine, PG: phosphatidylglycerol, PA: phosphatidic acid, and SM: sphingomyelin.

**Table 2. Altered peaks identified by MALDI-IMS and corresponding candidate phospholipids in negative ion mode.**

**A.** Upregulated in Tumor

| Case 1 | | | | | Case 2 | | | | |
|---|---|---|---|---|---|---|---|---|---|
| m/z | ratio[a] | Candidate phospholipids | | | m/z | ratio[a] | Candidate phospholipids | | |
| 809.48 | 9.7 | PI (32:0) | | | 661.46 | 3.1 | PA(33:0) | | |
| 630.46 | 3.5 | PE(28:2) | | | 659.20 | 2.9 | - | | |
| 661.46 | 3.4 | PA(33:0) | | | 630.46 | 2.8 | PE(28:2) | | |
| 642.46 | 3.3 | - | | | 644.48 | 2.6 | PE(30:1) * | | |
| 659.16 | 3.2 | - | | | 631.47 | 2.5 | - | | |
| 810.50 | 3.0 | PE(42:10) | PS(38:4) | | 647.45 | 2.4 | PA(32:0) | | |
| 616.45 | 2.5 | - | | | 642.46 | 2.3 | - | | |
| 837.53 | 2.0 | PI(34:0) | | | 643.47 | 2.2 | PA(32:2) | | |
| | | - | | | 616.45 | 2.1 | - | | |
| | | | | | 660.20 | 2.0 | - | | |
| | | | | | 718.51 | 2.0 | PC(31:0) | PE(34:0) | PS(31:1) |

**B.** Downregulated in Tumor

| Case 1 | | | | | Case 2 | | | | |
|---|---|---|---|---|---|---|---|---|---|
| m/z | ratio[a] | Candidate phospholipids | | | m/z | ratio[a] | Candidate phospholipids | | |
| 786.50 | 0.3 | PE(40:8) | PS(36:2) | | 695.45 | 0.2 | PA(36:4) | | |
| 778.51 | 0.3 | PE(40:7) | PS(36:1) | PC(38:6) | 696.45 | 0.3 | PE(33:4) | | |
| 862.52 | 0.3 | PS(42:6) | | | 859.51 | 0.3 | PI (36:3) | | |
| 861.52 | 0.3 | PI(36:2) | | | 786.50 | 0.3 | PE(40:8) | PS(36:2) | |
| 776.50 | 0.3 | PC(36:6) | PS(36:7) | PS(35:0) | 835.50 | 0.3 | PI (34:1) | | |
| 728.52 | 0.4 | PC(32:2) | PE(37:2) | PE(36:1) * | 836.52 | 0.3 | PE(44:11) | PS(40:5) | |
| | | PC(33:1)* | | | 770.51 | 0.3 | PS(35:3) | | |
| 771.51 | 0.4 | PA(42:8) | PG(36:3) | | 834.48 | 0.4 | PE(44:12) | PS(40:6) | |
| 770.51 | 0.4 | PS(35:3) | | | 774.52 | 0.4 | PC(36:7) | PS(35:1) | PE(40:6)* |
| 772.51 | 0.4 | PS(35:2) | PC(36:8) | PE(40:7) * | 747.47 | 0.4 | PA(40:6) | PG(34:1) | |
| 773.51 | 0.4 | PA(42:7) | | | 790.53 | 0.4 | PC(37:6) | PE(40:6) | PS(36:0) |
| 885.53 | 0.5 | PI (36:4) | | | 861.52 | 0.4 | PI (36:2) | | |
| 886.52 | 0.5 | PS(44:8) | | | 883.49 | 0.4 | PI (38:5) | | |
| 727.49 | 0.5 | PA(38:2) | | | 862.52 | 0.4 | PS(42:6) | | |
| 726.50 | 0.5 | PE(35:3) | PC(32:3) | PE(36:2) * | 884.51 | 0.4 | PS(44:9) | | |
| | | | | | 788.53 | 0.4 | PE(40:7) | PS(36:1) | PC(38:6) * |
| | | | | | 789.51 | 0.4 | - | | |
| | | | | | 721.47 | 0.4 | PA(38:5) | PG(32:0) | |
| | | | | | 863.52 | 0.4 | PI (36:1) | | |
| | | | | | 750.51 | 0.4 | PC(34:5) | PE(37:5) | PE(38:4) * |
| | | | | | 833.50 | 0.4 | PI (34:2) | | |
| | | | | | 764.48 | 0.5 | PE(38:5) | PC(35:5) | |
| | | | | | 697.45 | 0.5 | - | | |
| | | | | | 740.51 | 0.5 | PC(33:3) | PE(36:3) | |
| | | | | | 748.48 | 0.5 | PE(37:6) | PC(34:6) | PS(33:0) |
| | | | | | 857.48 | 0.5 | PI (36:4) | | |
| | | | | | 751.50 | 0.5 | PE(37:6) | | |

(*Continued*)

**Table 2.** (Continued)

| | | | | 794.51 | 0.5 | PS(37:5) | | |
|---|---|---|---|---|---|---|---|---|

The list of *m/z*, where the mean spectrum is significantly different between tumor and non-tumor regions (Welch's t-test, p < 0.01) and the ratio of mean spectral intensities is more than double, and the median value of spectral intensities is more than 200. The candidate phospholipids estimated from the Human Metabolome Database (https://hmdb.ca/spectra/ms/search) corresponding to these *m/z* values are presented. The *m/z* common to Case 1 and Case 2 is shown by shading.

[a] The ratio of relative signal intensity of tumor region to non-tumor region.

*deoxidation products. PC: phosphatidylcholine, PE: phosphatidylethanolamine, PI: phosphatidylinositol, PS: phosphatidylserine, PG: phosphatidylglycerol, PA: phosphatidic acid, and SM: sphingomyelin.

### Phospholipid candidates characteristic of human parotid cancer tissues

The candidate phospholipids corresponding to altered peaks identified by IMS were estimated from the database. The number of phospholipid candidates in the positive mode that were upregulated in tumor area were: 4, PC; 7, PE; 0, PI; 2, PS; 2, PG; 3, PA; and 2, SM; downregulated in tumor area were 11, PC; 9, PE; 0, PI; 5, PS; 0, PG; 10, PA; and 2, SM (Table 1). As for the negative mode, the number of phospholipid candidates that were upregulated in tumor area were: 1, PC; 4, PE; 2, PI; 2, PS; 0, PG; 2, PA; and 0, SM; downregulated in tumor area were 12, PC; 15, PE; 8, PI; 15, PS; 3, PG; 7, PA; and 0, SM (Table 2).

## Discussion

In this study, the lipid distribution in human parotid cancer tissues was analyzed using MALDI-IMS, which identified several candidate phospholipids whose expression levels differed between tumor and non-tumor regions. This suggests that parotid cancer tissue has a markedly different phospholipid composition compared to that of non-tumor tissue. Alterations in the lipid composition of tissues have been reported in several cancers, including breast cancer [18, 21], prostate cancer [17, 19], lung cancer [22], kidney cancer [23], pharyngeal cancer [24], and oral cancer [20, 25, 26]. Furthermore, it has been suggested that these specific lipid profiles vary from among carcinomas, and may be potential diagnostic, prognostic, and predictive biomarkers. These facts suggest that it is necessary to individually examine each tumor. Therefore, it is worthwhile to further investigate the changes in the lipid composition in parotid cancer.

Molecular visualization has traditionally been important in the characterization of clinical specimens of parotid cancer. For example, immunohistochemical staining was performed for visualization of the expression of protein of myoepithelial and basal cell markers, such as alpha-smooth muscle actin, calponin, p63, and S100 [27, 28], and receptors, such as HER2 [29] and androgen receptors. In addition, fluorescence in situ hybridization is used for the detection of fusion genes, such as the ETV6-NTRK3 fusion gene in secretory carcinoma and the CRTC1/3-MAML2 fusion gene in mucoepidermoid carcinoma [3, 30]. In this study, MALDI-IMS was used to visualize lipid molecules in clinical parotid cancer specimens. MALDI-IMS has been used to investigate the localization of phospholipids in some cancer tissues [15], as the information from mass spectrometry, together with their location in the sample, can be obtained simultaneously. The results of this study show for the first time that MALDI-IMS may be useful as a new method for the clinical study of parotid cancer.

Here, MALDI-IMS analysis was performed in both positive and negative ion modes to comprehensively evaluate a large number of molecules. It is known that the positive ion mode is excellent for detecting PC and SM, and the negative ion mode is suitable for detecting PI, PS, and PG [31]; PE is detected in both modes [32]. In this study, there was a tendency to

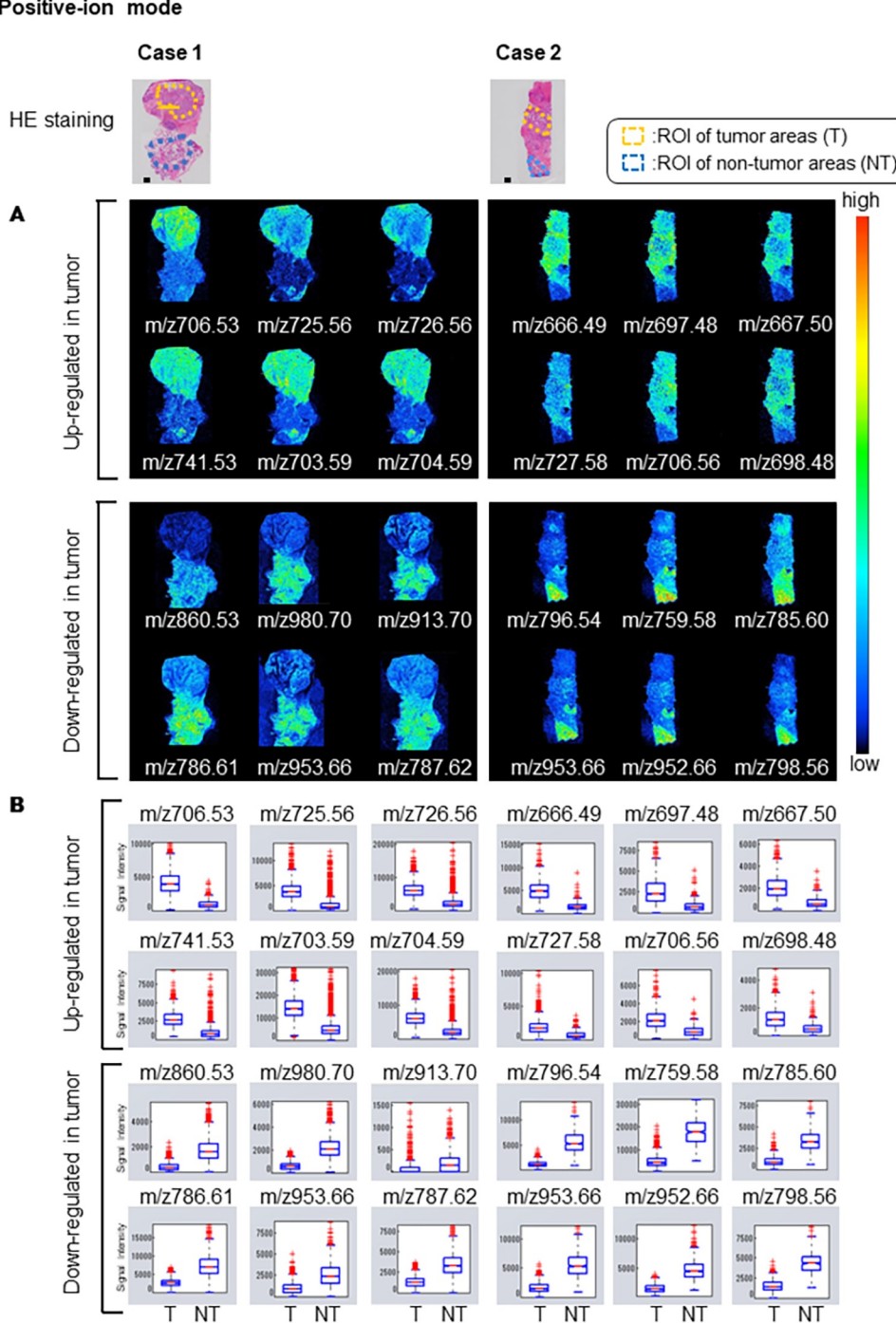

**Fig 1. Visualization by positive ion mode IMS of molecular distributions.** Hematoxylin and eosin (HE) stained images show defined regions of interest (ROIs) of tumor (T) and non-tumor (NT) areas. A. Representative images of peaks (*m/z*) with significantly different signal intensities in the tumor and non-tumor areas, which are indicated by red and blue circles on the tumor/non-tumor dot graph in S2 Fig. The threshold of the color scale was adjusted for each ion image to show a clear distribution. B. Box plots represent the signal intensities of the T and NT areas at each *m/z*. The significance of the difference of signal intensity between tumor and non-tumor areas was determined using Welch's t-test. A p-value of less than 0.01 was obtained for all *m/z* signals compared here.

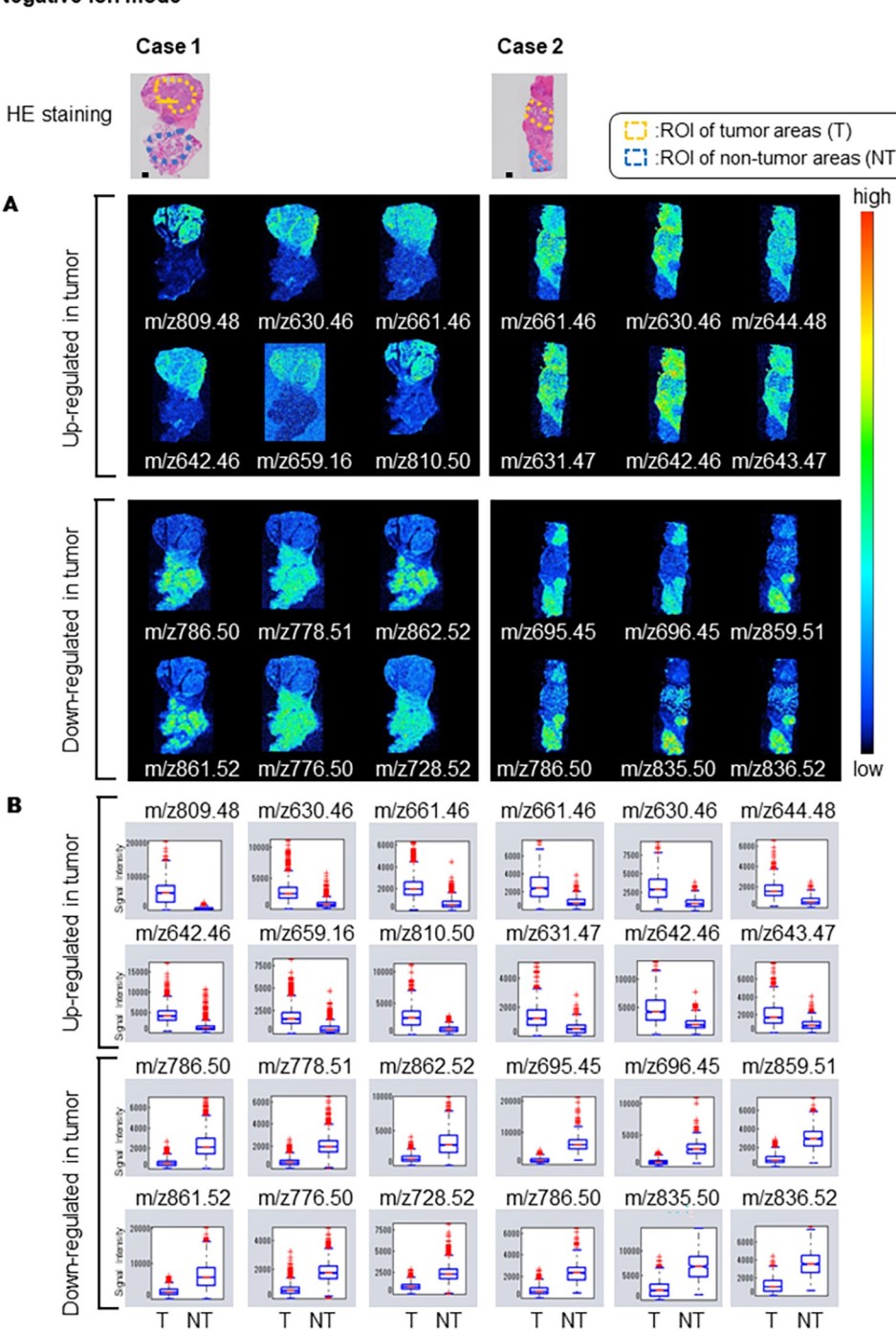

**Fig 2. Visualization by negative ion mode IMS of molecular distributions.** Hematoxylin and eosin (HE) stained images show defined tumor (T) and non-tumor (NT) areas. A. Representative images of peaks (*m/z*) with significantly different ion intensities in the tumor and non-tumor areas, which are indicated by red and blue circles on the tumor/non-tumor graph in S3 Fig. The threshold of the color scale was adjusted for each ion image to show a clear distribution. B. Box plots represent the signal intensities of the T and NT regions at each *m/z*. The significance of the difference between tumor and non-tumor regions was determined using Welch's t-test. A p-value of less than 0.01, was obtained for all *m/z* signals compared here.

detect more PC and SM in positive ion mode, while more PI and PG were detected in the negative ion mode as phospholipid candidates whose expression levels differed significantly between tumor and non-tumor regions. In addition, PI was not detected in the positive ion mode, and SM was not detected in the negative ion mode. These results are consistent with the characteristics of the analytical method and suggest that further detailed analyses should be carried out, considering the characteristics of the analytical method. Furthermore, in this study, we used 9-aminoacridine as a matrix, which is one of the most commonly used MALDI matrices for lipid analysis, because it can ionize target molecules in both positive and negative ion modes [33]. Since the analytical sensitivity of lipid imaging depends on the type of matrix [34], the choice of matrix should also be considered in future analyses.

The present study suggests that phospholipid composition is significantly different between tumor and non-tumor regions of parotid cancer. The differences in the profiles of these phospholipids may be related to the well-known features of cancer cells. In general, rapidly proliferating cancer cells are characterized by enhanced lipid synthesis and changes in the composition of the cell membrane, which may be associated with an increase in PC and saturated phospholipids [8, 9]. The number of lipid rafts involved in cell signaling is increased in cancer cells, which may be associated with increased cholesterol and SM [35]. In addition, lipids, such as free fatty acids and prostaglandins, are involved in the communication between cancer and stromal cells. Therefore, it is possible that these changes in the cancer cells of the parotid cancer lesions in the present study were due to differences in lipid profiles. The phospholipid candidates common to the two cases of different histological types in this study are likely to be related to the characteristics of the cancer cells. In contrast, for phospholipid candidates whose expression is upregulated in a particular tissue type, it may be characteristic of that tissue type. Since the present study was based only on two cases of parotid cancer, further analysis with a larger number of cases is needed to clarify these possibilities.

Although the present results have revealed novel insights into the distribution of lipids in parotid cancer, this study should also be considered in the context of certain limitations. First, the design of this study aimed to confirm the distribution of the compound in the tissue and not to identify the compound. In this study, we estimated candidate phospholipids from a database based on mass peaks of compounds differentially expressed in tumor and non-tumor areas of parotid cancer tissues. We believe that other methods, such as LC MS/MS analysis [36] of parotid cancer tissue, should be used to identify lipid molecules that are differentially expressed in tumor and non-tumor tissues. Therefore, we plan to perform additional experiments and report those findings in a separate study. Second, only two cases of parotid cancer were included in this study, which is not sufficient to clarify the characteristics of parotid cancer. Since parotid cancer has diverse clinical and histological characteristics, such as age and sex, it is necessary to analyze a larger number of cases in order to clarify the effects of these factors on phospholipid expression. Nevertheless, this is a preliminary pilot study with a limited sample size, and the main aim was to demonstrate that the lipid profile of parotid cancer tissue is a characteristic of the tumor lesion. In this context, this study is a promising proof of concept and opens up a new pathway for the use of lipid profiles as a potential diagnostic tool for parotid cancer. These limitations need to be addressed in future research.

## Conclusions

The lipid distribution in human parotid cancer tissues was analyzed using MALDI-IMS, and candidate phospholipids differentially expressed in tumor and non-tumor areas were profiled. Further investigation of changes in lipid metabolism in parotid cancer is worthwhile.

## Supporting information

**S1 Fig. Parotid gland cancer tissue.** Parotid gland cancer tissue removed during surgery and cut out tissue in Cases 1 (A) and 2 (B). Scale bar: 1.0 cm. HE-stained tissue of Cases 1 (C) and 2 (D). Scale bar: 500 μm; enlarged images: 50 μm. The yellow dashed area indicates ROI of tumor areas, and the blue dashed area indicates ROI of non-tumor areas. HE, hematoxylin and eosin; ROI, region of interest.
(TIF)

**S2 Fig. Mass spectra and relative intensity ratios of tumor and non-tumor areas in positive ion modes.** The spectra in the mass range of *m/z* 600–1000 in positive ion modes are shown for the tumor and non-tumor regions distinguished in S1 Fig of Case 1 and Case 2. The horizontal axis shows *m/z*, and the vertical axis shows the relative intensity. The dot graph of the ratio of relative intensities of tumor and non-tumor regions at each *m/z* is shown below. Of the peaks (*m/z*) with a median value of spectral intensity greater than 200, red circles indicate the top six *m/z* with a higher expression ratio and blue circles indicate the top six *m/z* with a lower expression ratio in tumor areas compared to non-tumor regions. Fig 1 shows the IMS images of each of these m/z values.
(TIF)

**S3 Fig. Mass spectra and relative intensity ratios of tumor and non-tumor areas in negative ion modes.** The spectra in the mass range of *m/z* 600–1000 in negative ion modes are shown for the tumor and non-tumor regions distinguished in S1 Fig of Case 1 and Case 2. The horizontal axis shows *m/z*, and the vertical axis shows the relative intensity. The dot graph of the ratio of relative intensities of tumor and non-tumor regions at each *m/z* is shown below. Of the peaks (*m/z*) with the median value of spectral intensity greater than 200, red circles indicate the top six *m/z* with a higher expression ratio and blue circles indicate the top six *m/z* with a lower expression ratio in tumor regions compared to non-tumor regions. Fig 2 shows the IMS images of each of these *m/z* values.
(TIF)

## Acknowledgments

We gratefully acknowledge the support of Shimazu, Inc. for IMS analysis in this research.

## Author Contributions

**Conceptualization:** Hirofumi Kanetake, Nahoko Kato-Kogoe.

**Data curation:** Hirofumi Kanetake, Nahoko Kato-Kogoe, Tetsuya Terada, Wataru Hamada, Yoichiro Nakajima.

**Formal analysis:** Hirofumi Kanetake, Nahoko Kato-Kogoe, Tetsuya Terada, Takaaki Ueno.

**Funding acquisition:** Hirofumi Kanetake, Nahoko Kato-Kogoe, Tetsuya Terada, Yoichiro Nakajima.

**Investigation:** Hirofumi Kanetake, Nahoko Kato-Kogoe, Yoshitaka Kurisu, Yoichiro Nakajima, Ryo Kawata.

**Methodology:** Hirofumi Kanetake, Nahoko Kato-Kogoe, Yoichiro Nakajima.

**Project administration:** Hirofumi Kanetake, Wataru Hamada, Takaaki Ueno, Ryo Kawata.

**Resources:** Hirofumi Kanetake, Wataru Hamada.

**Software:** Hirofumi Kanetake.

**Supervision:** Yoshinobu Hirose, Takaaki Ueno, Ryo Kawata.

**Validation:** Nahoko Kato-Kogoe.

**Visualization:** Nahoko Kato-Kogoe.

**Writing – original draft:** Hirofumi Kanetake.

**Writing – review & editing:** Hirofumi Kanetake, Nahoko Kato-Kogoe, Yoichiro Nakajima.

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
