## [Decision Letter · Decision Letter 0]

6 Oct 2021

PONE-D-21-27808Distribution of phospholipids in parotid cancer by matrix-assisted laser desorption/ionization imaging mass spectrometryPLOS ONE

Dear Dr. Kanetake,

Thank you for submitting your manuscript to PLOS ONE. After careful consideration, we feel that it has merit but does not fully meet PLOS ONE’s publication criteria as it currently stands. Therefore, we invite you to submit a revised version of the manuscript that addresses the points raised during the review process.

We look forward to receiving your revised manuscript.

Kind regards,

Joseph Banoub, Ph,D., D. Sc.,FCIC,FRSC

Academic Editor

PLOS ONE

Journal Requirements:

 [NO: The funders had no role in study design, data collection and analysis, decision to publish, or preparation of the manuscript.]

[NO authors have competing interests]. 

4. Thank you for stating the following in the Funding Section of your manuscript: 

[This research was partially supported by the Private University Research Branding Project (2017-2019) of the Ministry of Education, Culture, Sports, Science]

 [NO: The funders had no role in study design, data collection and analysis, decision to publish, or preparation of the manuscript.]

Additional Editor Comments:

I am pleased to inform you that PLOS ONE has found that your manuscript entitled: " Distribution of phospholipids in parotid cancer by matrix-assisted laser desorption/ionization imaging mass spectrometry; Manuscript Number PONE-D-21-27808" worth publishing.

However, according to the one referee, your manuscript needs some corrections and needs major revision

Reviewers' comments:

Reviewer's Responses to Questions

**Comments to the Author**

1. Is the manuscript technically sound, and do the data support the conclusions?

Reviewer #1: Partly

Reviewer #2: Yes

2. Has the statistical analysis been performed appropriately and rigorously? 

Reviewer #1: I Don't Know

Reviewer #2: I Don't Know

3. Have the authors made all data underlying the findings in their manuscript fully available?

Reviewer #1: Yes

Reviewer #2: Yes

4. Is the manuscript presented in an intelligible fashion and written in standard English?

Reviewer #1: Yes

Reviewer #2: Yes

5. Review Comments to the Author

Reviewer #1: Major comment:

The paper tackles and interesting subject and uses IMS to profile lipids for Parotid cancer, an absolutely novel idea. However, I suggest that the paper is re-written and significantly shorten as it is more suitable for a “short report” rather than being a full research article. The authors rightfully acknowledge the main limitation of the study that is the sample size and with only 2 samples for rather different cancer type, the paper is only suitable for a short report or short communication-type paper. It will require significant reduction and re-writing

Additional comments:

• The instrument type needs to be described including its resolution and mass error. I see m/z values reported with one significant digit. That is not suitable for discovery-based paper like this one.

• Putative identification is needed; hence the importance of the mass accuracy

• Why the authors did not do MSMS

• What normalization strategy was used – did they use matrix peaks for that

• The two patients and males within very different age groups- is this a concern?

• Why 9-aminoacridine is used for both positive and negative mode analysis. Matrix choice will have a profound effect on IMS results

Reviewer #2: PONE-D-21-27808

This work focuses on the identification of phospholipids biomarkers for parotid cancer. These biomarkers were identified by comparing the lipid profiles of the tumor and non-tumor tissue by MALDI imaging. Consequently, several lipid peaks were significantly regulated (down- or Up-regulated) due to the parotid cancer. This pilot study is a straightforward biomarker identification using comparative lipidomics, and it opens a way for further future studies on the alterations in lipid metabolism of parotid cancer.

Overall, I recommend the publication of this work. Also, I recommend adding a separate conclusion section at the end.

6. PLOS authors have the option to publish the peer review history of their article (what does this mean?). If published, this will include your full peer review and any attached files.

Reviewer #1: No

Reviewer #2: No

---

## [Author Response · Author response to Decision Letter 0]

13 Nov 2021

November 12, 2021.

Dr. Joseph Banoub

Academic Editor,

PLOS ONE

Dear Dr. Joseph Banoub,

Thank you for your comments regarding our paper (Manuscript No: Ms. PONE-D-21-27808) entitled “Distribution of phospholipids in parotid cancer by matrix-assisted laser desorption/ionization imaging mass spectrometry” by Kanetake H. et al. for publication in PLOS ONE. We have modified our paper according to the reviewers’ comments. Our responses to each point raised by reviewers are as follows:

Reply to reviewer #1:

1. Concern of the reviewer: Major comment: The paper tackles and interesting subject and uses IMS to profile lipids for Parotid cancer, an absolutely novel idea. However, I suggest that the paper is re-written and significantly shorten as it is more suitable for a “short report” rather than being a full research article. The authors rightfully acknowledge the main limitation of the study that is the sample size and with only 2 samples for rather different cancer type, the paper is only suitable for a short report or short communication-type paper. It will require significant reduction and re-writing

Our response: Thank you for the time you spent reviewing our work. Because of the relatively low incidence of parotid cancer, it takes years to perform studies with large sample sizes. Therefore, as the reviewer points out, it is worth reporting the results of this sample size as a "short report". Accordingly, we have shortened the manuscript by 294 words to a revised total of 2615 words. This has been achieved by changing the style of expression and deleting unnecessary text. In addition, three Figures have been adapted to supplementary materials. Furthermore, the title has been revised. Please refer to these in the revised version.

2. Concern of the reviewer: Additional comments: The instrument type needs to be described including its resolution and mass error. I see m/z values reported with one significant digit. That is not suitable for discovery-based paper like this one.

Our response: As the reviewer pointed out, the resolution and mass accuracy of the mass spectrometer are important in the interpretation of the results. Therefore, the specifications of the imaging mass microscope (iMScope TRIO) used in this study, with a mass resolution of 10,000 and a mass accuracy of less than 20 ppm, have been added to the Materials and methods section (p.7, lines 126–127). Additionally, in accordance with the reviewers' instructions regarding the notation of m/z values, all m/z values have been corrected to two significant digits.

Revised text: p.7, lines 126–127; The tissue sections were analyzed using an imaging mass microscope (iMScope TRIO, Shimadzu Corporation, Kyoto, Japan) equipped with a 355-nm Nd: YAG laser, which has a mass resolution of 10,000 and a mass accuracy of less than 20 ppm. 

3. Concern of the reviewer: Additional comments: Putative identification is needed; hence the importance of the mass accuracy.

Our response: As indicated by the reviewer, we also think that the putative identification of compounds would be valuable. In this study, we estimated candidate phospholipids from a database based on mass peaks of compounds differentially expressed in tumor and non-tumor areas of parotid cancer tissues. For more accurate identification of candidate phospholipids, other experiments, such as MS/MS, are required. However, the present study aimed to determine the distribution of the phospholipids in parotid cancer tissue, and we consider this paper to be a report of a preliminary study. The results of this study suggest that there may be phospholipids with different distribution patterns in tumor and non-tumor areas, which will be identified in more detail and presented in a separate study. To make this point clearer, we have added the following statements to the Discussion section.

Revised text: p.17, line 324–p.18, line 331; First, the design of this study aimed to confirm the distribution of the compound in the tissue and not to identify the compound. In this study, we estimated candidate phospholipids from a database based on mass peaks of compounds differentially expressed in tumor and non-tumor areas of parotid cancer tissues. We believe that other methods, such as LC MS/MS analysis [34] of parotid cancer tissue, should be used to identify lipid molecules that are differentially expressed in tumor and non-tumor tissues. Therefore, we plan to perform additional experiments and report those findings in a separate study.

4. Concern of the reviewer: Additional comments: Why the authors did not do MSMS.

Our response: The remarks of the reviewer are very valuable. We have included additional details on this issue in the “Discussion” section (p.17, line 324–p.18, line 331). As described above, the present study focuses on investigating the distributional localization of phospholipid expression in parotid cancer tissues to determine whether there are phospholipids differentially expressed between tumor and non-tumor areas. Since the results of this study suggest this possibility, the next step should be the identification of candidate phospholipids. In this study, the candidate phospholipids were estimated from the database based on the peak mass by IMS. As the reviewer pointed out, MS/MS is very valuable for validation and identification of the obtained presumed phospholipids. However, in the present case, it was difficult to collect appropriate samples from tumor and non-tumor areas for MS/MS analysis. Furthermore, parotid cancer is a rare disease, and it is not easy to collect a large number of cases. Therefore, as a next step, we plan to increase the number of parotid carcinoma cases, identify phospholipids that are differentially expressed between tumor and non-tumor areas, and report those results in a separate study in the near future.

Revised text: p.17, line 324–p.18, line 331; First, the design of this study aimed to confirm the distribution of the compound in the tissue and not to identify the compound. In this study, we estimated candidate phospholipids from a database based on mass peaks of compounds differentially expressed in tumor and non-tumor areas of parotid cancer tissues. We believe that other methods, such as LC MS/MS analysis [34] of parotid cancer tissue, should be used to identify lipid molecules that are differentially expressed in tumor and non-tumor tissues. Therefore, we plan to perform additional experiments and report those findings in a separate study.

5. Concern of the reviewer: Additional comments: What normalization strategy was used – did they use matrix peaks for that.

Our response: As described in the Materials and methods section (p.7, lines 134–136), in IMS data analysis, the acquired raw mass spectra were normalized by the total ion current (TIC) using Imaging MS Solution version 1.12.26 (Shimadzu Corporation). The matrix peaks were not used specifically for normalization. The variation in the ionization efficiency, which is caused by the heterogeneous distribution of matrix crystals and their sublimation during measurement, was eliminated for each data point by equalizing the TIC of each mass spectra. Normalization of spectra by TIC has been reported to improve IMS visualization quality (Sugiura Y, et al. J Lipid Res. 2009).

6. Concern of the reviewer: Additional comments: The two patients and males within very different age groups- is this a concern?

Our response: The two cases in this study are a 32-year-old man with acinic cell carcinoma and a 65-year-old man with mucoepidermoid carcinoma. Parotid gland cancer is a relatively rare lesion with a wide variation in biological and histological characteristics. Epidemiologically, there is a wide age range at diagnosis of parotid cancer. Therefore, this preliminarily study included acinic cell carcinoma and mucoepidermoid carcinoma, which are common histological types of parotid cancer, at different ages. Based on the results of this study, it is not possible to evaluate the results according to age, gender, or histological types. Further analysis of a larger number of cases will help to clarify the influence of these factors on phospholipid expression. The Discussion section (p.18, lines 331–335) has been revised to make these clearer.

Revised text: p.18, lines 331–335; Second, only two cases of parotid cancer were included in this study, which is not sufficient to clarify the characteristics of parotid cancer. In addition, because there are many histological types of parotid cancer, a larger number of cases should be analyzed to clarify the differences between each histological type. Since parotid cancer has diverse clinical and histological characteristics, such as age and sex, it is necessary to analyze a larger number of cases in order to clarify the effects of these factors on phospholipid expression.

7. Concern of the reviewer: Additional comments: Why 9-aminoacridine is used for both positive and negative mode analysis. Matrix choice will have a profound effect on IMS results. 

Our response: 9-aminoacridine is one of the MALDI matrices commonly used for lipid analysis. The aim of this study was to comprehensively detect phospholipid compounds differentially expressed between tumor and non-tumor areas. Thus, we performed measurements in both modes since phospholipid compounds that are likely to be detected in positive and negative modes are different. 9-aminoacridine can ionize target molecules in both positive and negative ion modes and has been shown to be useful in the analysis of lipid imaging (Perry WJ, et al. J Mass Spectrom. 2020). Therefore, in the present study, 9-aminoacridine was used as a matrix in the first stage of IMS of phospholipids in parotid cancer. As pointed out by the reviewer, the analytical sensitivity of lipid imaging can vary when different types of matrices are employed (Tobias F, et al. J Proteome Res. 2020); thus, the choice of matrix should be considered a future issue. We have included additional details on this issue in the “Discussion” section (p. 16, lines 301–p.17. lines 305 ).

Revised text: p. 16, lines 301–p.17. lines 305; Furthermore, in this study, we used 9-aminoacridine as a matrix, which is one of the most commonly used MALDI matrices for lipid analysis, because it can ionize target molecules in both positive and negative ion modes (Perry WJ, et al. J Mass Spectrom. 2020). Since the analytical sensitivity of lipid imaging depends on the type of matrix (Tobias F, et al. J Proteome Res. 2020), the choice of matrix should also be considered in future analyses.

Reply to reviewer #2:

1. Concern of the reviewer: This work focuses on the identification of phospholipids biomarkers for parotid cancer. These biomarkers were identified by comparing the lipid profiles of the tumor and non-tumor tissue by MALDI imaging. Consequently, several lipid peaks were significantly regulated (down- or Up-regulated) due to the parotid cancer. This pilot study is a straightforward biomarker identification using comparative lipidomics, and it opens a way for further future studies on the alterations in lipid metabolism of parotid cancer.

Overall, I recommend the publication of this work. Also, I recommend adding a separate conclusion section at the end.

Our response: Thank you very much for the time you spent reviewing our work. In accordance with the reviewer's comment, we have added “Conclusions” section in the revised manuscript. 

Revised text: p. 18, lines 342–346; Conclusions: The lipid distribution in human parotid cancer tissues was analyzed using MALDI-IMS, and candidate phospholipids differentially expressed in tumor and non-tumor areas were profiled. Further investigation of changes in lipid metabolism in parotid cancer is worthwhile.

Once again, we thank you for your kind suggestions and the time you have spent reviewing our work. We hope that you will find our revised manuscript suitable for publication in PLOS ONE.

Yours sincerely,

Hirofumi Kanetake 

Department of Otorhinolaryngology-Head and Neck Surgery, 

Osaka Medical and Pharmaceutical University.

E-mail: hirofumi.kanetake@ompu.ac.jp.

---

## [Editor Report · Decision Letter 1]

3 Dec 2021

Short communication: Distribution of phospholipids in parotid cancer by matrix-assisted laser desorption/ionization imaging mass spectrometry

PONE-D-21-27808R1

Dear Dr. Kanetake,

We’re pleased to inform you that your manuscript has been judged scientifically suitable for publication and will be formally accepted for publication once it meets all outstanding technical requirements.

Kind regards,

Joseph Banoub, Ph,D., D. Sc., FCIC, FRSC

Academic Editor

PLOS ONE

Additional Editor Comments (optional):ALL

All queries  have been  answered
---

## [Editor Report · Acceptance letter]

9 Dec 2021

PONE-D-21-27808R1 

Short communication: Distribution of phospholipids in parotid cancer by matrix-assisted laser desorption/ionization imaging mass spectrometry 

Dear Dr. Kanetake:

I'm pleased to inform you that your manuscript has been deemed suitable for publication in PLOS ONE. Congratulations! Your manuscript is now with our production department. 

Kind regards, 

on behalf of

Dr. Joseph Banoub 

Academic Editor

PLOS ONE